# Serum Uric Acid Levels in Parkinson’s Disease: A Cross-Sectional Electronic Medical Record Database Study from a Tertiary Referral Centre in Romania

**DOI:** 10.3390/medicina58020245

**Published:** 2022-02-06

**Authors:** Adela Dănău, Laura Dumitrescu, Antonia Lefter, Bogdan Ovidiu Popescu

**Affiliations:** 1Department of Clinical Neurosciences, Colentina Clinical Hospital Neurology Division, Carol Davila University of Medicine and Pharmacy, 020021 Bucharest, Romania; adela.danau@drd.umfcd.ro (A.D.); laura.dumitrescu@drd.umfcd.ro (L.D.); antonia.lefter@rez.umfcd.ro (A.L.); 2Laboratory of Neurosciences and Experimental Myology, Victor Babeș National Institute of Pathology, 050096 Bucharest, Romania

**Keywords:** antioxidant, Parkinson’s disease, prophylaxis, therapy, uric acid

## Abstract

*Background and Objectives*: Parkinson’s disease (PD) is a prevalent neurodegenerative condition responsible for progressive motor and non-motor symptoms. Currently, no prophylactic or disease-modifying interventions are available. Uric acid (UA) is a potent endogenous antioxidant, resulting from purine metabolism. It is responsible for about half of the antioxidant capacity of the plasma. Increasing evidence suggests that lower serum UA levels are associated with an increased risk of developing PD and with faster disease progression. *Materials and Methods*: We conducted an electronic medical record database study to investigate the associations between UA levels and different characteristics of PD. *Results*: Out of 274 datasets from distinct patients with PD, 49 complied with the predefined inclusion and exclusion criteria. Lower UA levels were significantly associated with the severity of parkinsonism according to the Hoehn and Yahr stage (r_s_ = 0.488, *p* = 0.002), with the motor complications of long-term dopaminergic treatment (r = 0.333, *p* = 0.027), and with the presence of neurocognitive impairment (r = 0.346, *p* = 0.021). *Conclusions*: Oxidative stress is considered a key player in the etiopathogenesis of PD, therefore the involvement of lower UA levels in the development and progression of PD is plausible. Data on the potential therapeutic roles of elevating serum UA (e.g., by precursor administration or diet manipulation) are scarce, but considering the accumulating epidemiological evidence, the topic warrants further research.

## 1. Introduction

Parkinson’s disease (PD) is a neurodegenerative condition affecting over 6 million people worldwide [1]. Parkinsonism, the motor aspect of the disease, is caused by the degeneration of the dopaminergic neurons in the substantia nigra pars compacta [2]. It consists of bradykinesia, with resting tremor and/or rigidity, accompanied by freezing and postural instability [3]. It is asymmetrical at onset and responsive to levodopa [2,3]. Gait impairment and important overall motor disability develop as the disease progresses [1,3]. Disability is also caused by the coexistence and progression of non-motor symptoms, which may precede the onset of parkinsonism by several years [3]. These nonmotor symptoms range from hyposmia, neurocognitive impairment, and dysautonomia (e.g., chronic constipation, orthostatic hypotension, etc.), to rapid eye movement sleep behavior disorder, sensitive symptoms, and olfactory hallucinations [4,5].

Most cases of PD are multifactorial and sporadic, but monogenic forms, familial or de novo, with onset at younger ages, also exist [6]. The etiopathology of sporadic cases is complex and still incompletely understood, with several environmental factors playing protective or deleterious roles in people with a genetic predisposition for PD [3]. Mounting evidence suggests that the pathology of PD begins in the gut and/or in the olfactory mucosa [7], spreading along the vagus and olfactory nerves, and entering the brain via the dorsal nucleus of the vagus and the olfactory bulb [8]. Prophylactic or disease-modifying treatments are not yet available [1]. Uric acid (UA) and its anionic salt, urate (U), are potent endogenous antioxidants that result from the metabolism of exogenous or endogenous purines, and account for about half of the antioxidant capacity of the serum [9,10]. Oxidative stress results from the imbalance between the production and the elimination of reactive oxygen species, leading to oxidation reactions. Excessive oxidative stress is considered a key factor in the development and progression of PD [11], and may cause mitochondrial damage, excitotoxicity, and neuronal death. It might also partly explain the selectivity of the neurodegenerative processes in PD, certain neurons, such as the dopaminergic neurons in the substantia nigra, being more prone or more susceptible to oxidative stress [12,13]. Moreover, the pathological hallmark of PD is alpha-synuclein misfolding, with intraneuronal accumulation of insoluble aggregates, namely Lewy bodies and Lewy neuritis, with neurodegeneration [14]. In this respect, a recent experimental study found that oxidative stress increases the production of oxidatively-modified alpha-synuclein, as well as the pathological aggregation of alpha-synuclein and neuronal loss [15].

Serum UA levels above the upper reference range (i.e., hyperuricemia) are associated with gout [16], arterial hypertension [17,18], and metabolic syndrome [19], among others, and have a potentially negative impact on cortical thickness [16] and on the cerebrovascular status [20]. There is epidemiological evidence that lower levels of serum UA and U are associated with an increased risk of PD; this was shown in two large prospective studies [21,22], four case–control studies [23,24,25,26], and a cross-sectional study [27]. Lower serum UA and U levels also associate with lower whole brain gray matter volume [28], and impaired connectivity of the substantia nigra [12]. Moreover, higher levels of serum UA were found to be associated with a significantly lower risk of PD, a dose–effect relation being observed [22]. Higher serum U levels were also found to be protective for PD in a case–control study [24], a bi-ethnic cohort study [29], and a cross-sectional survey [30]. In addition, the use of U-lowering drugs, considered a proxy indicator of high serum U levels, was associated with a lower risk of PD [31]. However, these associations may be due to a common genetic predisposition, and not to a direct association, as genetic variability, which influences serum UA levels, might also influence the risk of PD [32,33,34]. Furthermore, dietary factors known to lower serum UA levels, such as increased daily consumption of dairies, are associated with an increased risk of PD, and increased neuronal death [35,36], while factors known to elevate serum UA may be protective [37]. Data on the association between gout and the risk of PD are conflicting; for example, a recent large U.S. Medicare data study showed an increased risk of PD in people with gout [38], while two other large population-based studies [39,40] and a meta-analysis [41], found there is no effect, or even an inverse association, suggesting a potential protective effect of gout [31,42]. Epidemiologically, the potential protective effect of gout may be limited to men [43], although higher serum UA levels coexisting with preserved striatal dopamine activity were also found in women, suggesting that UA may have a neuroprotective effect in women [44].

Here, we report the results of a small cross-sectional observational study on the associations between serum UA levels and the characteristics of sporadic PD, performed on an electronic medical record database from a tertiary referral center in Romania. This work adds to the evidence that UA may play a role in the progression of motor and nonmotor symptoms in PD. The potential therapeutic role of UA in preventing PD or delaying its progression are also discussed.

## 2. Materials and Methods

We conducted a cross-sectional study using the electronic medical record database at Colentina Clinical Hospital, a tertiary referral center for PD in Bucharest, Romania. Our study invested the potential associations between serum UA levels and demographic and clinical characteristics of patients with PD. The study was performed in agreement with the Declaration of Helsinki and was approved by the local Ethics Committee (15/ 7 June 2021). All personal data were anonymized, according to the General Data Protection Regulation (2016/679). The study was conducted as part of the doctoral research of the first author (supervised by B.O.P.).

We screened all the available electronic medical records (i.e., from March 2018 until June 2021), using the Hippocrate Electronic Medical Record software, which is the electronic medical record database introduced in 2018 at Colentina Clinical Hospital. Study enrollment was based on predefined inclusion and exclusion criteria as follows: we included patients with a diagnosis of sporadic PD at discharge (i.e., ICD-10 code: G20), made by a neurologist working at Colentina Clinical Hospital; in order to be included, the patients must have had the onset of motor manifestations over the age of 50 years, no family history suggesting monogenic PD, and available results of a blood workup, including serum uric acid levels. We excluded patients with concurrent infectious diseases (including COVID-19), systemic inflammatory syndrome, haematological or oncological diseases, and patients with concurrent medication that may alter serum UA levels. We also excluded patients whose datasets were insufficient. If a patient had several datasets/hospital admissions complying with the enrolment criteria, we used the data from the initial evaluation (i.e., closer to the onset of the disease).

We collected the following data: sex, age, living environment (urban/rural), diagnosis of PD and of other diseases, medication, PD characteristics (time since onset of parkinsonism, severity of parkinsonism measured using the Hoehn and Yahr stage, presence of motor complications of long-term dopaminergic treatment, namely motor fluctuations and dyskinesia, and presence of neurocognitive impairment—documented in the medical records either as mild cognitive impairment, or as dementia/major neurocognitive impairment), and the serum UA levels (laboratory method: spectrophotometry; reference range: 3.4–7 mg/dL).

For statistical analysis, we used the IBM^®^ Statistical Package for Social Sciences^®^ (SPSS^®^) Statistics Subscription software. For the continuous variables, we calculated the means and standard deviations (SD). The statistical analysis included the Spearman correlation coefficient (r_s_), for ordinal and nominal categorical variables, and the Pearson coefficient with point-biserial correlation, for binary/dichotomous variables (r). We also performed logistic regression with odds ratio (OR) and 95% confidence interval (95% CI), using the serum UA level as the independent variable, and the motor complications and cognitive impairment as dependent variables. We considered the statistical significance threshold at *p* < 0.05. Normal distribution of data was assessed using graphical and numerical methods in SPSS^®^, including the Shapiro–Wilk test and the Kolmogorov–Smirnov test.

## 3. Results

The inclusion process, detailed in Figure 1, identified 274 patients diagnosed with PD, of whom 49 patients (33 males, 16 females) met all the inclusion criteria and had no exclusion criteria. The mean age was 69.1 ± 2.4 years (SD 8.6). The mean disease duration since the onset of parkinsonism was 5.1 ± 1.8 years (SD 5.2). Regarding the severity of the motor manifestations, stage 2 Hoehn and Yahr was the most prevalent, present in 34% of the patients (*n* = 17), followed by stage 3 in 26.5% of the patients (*n* = 13). Motor complications were present in 33.3% of the patients (*n* = 15), neurocognitive impairment was found in 32.7% (*n* = 16), and cerebrovascular disease in 26.5% (*n* = 13). For more details on the demographic and clinical characteristics of the study population, please see Table 1, and Appendix A.

The mean serum UA levels of the PD patients was 4.99 mg/dL ± 0.35 (SD 1.27), reference range 3.4–7 mg/dL. Serum UA levels were below the lower reference range in 12.2% of the PD patients (*n* = 6), and above the upper reference range in 6.1% of them (*n* = 3). We found significant associations between lower serum UA levels and the presence of motor complications (r = 0.333, *p* = 0.027), the presence of neurocognitive impairment (r = 0.346, *p* = 0.021), and the severity of the parkinsonism, assessed through the Hoehn and Yahr scale (r_s_ = 0.488, *p* = 0.002)—see Table 2 and Appendix A. Serum UA levels were also significantly associated with the male sex (*p* = 0.008). We found no association between sex and the Hoehn and Yahr stage, motor complications, or neurocognitive impairment. Logistic regression found significant associations for motor complications (OR = 0.552, 95% CI = 0.310–0.957, *p* = 0.034) and neurocognitive impairment (OR = 0.539, 95% CI = 0.310–0.937, *p* = 0.028). The estimated power of the study was 100% for the Hoehn and Yahr stage, 70% for the motor complications, and 50% for the neurocognitive impairment. No significant associations were found with disease duration, cerebrovascular disease, comorbidities, or concomitant medication (i.e., levodopa with carbidopa or with carbidopa and entacapone, dopamine agonists, or monoaminoxidase inhibitor).

## 4. Discussion

PD is a prevalent neurodegenerative disease for which prophylactic or disease-modifying treatments are not yet available [1]. Oxidative stress plays important roles in PD, especially in the dopaminergic neurons of the substantia nigra, whose degeneration is responsible for the development and progression of parkinsonism [14]. As discussed above, accumulating evidence suggests that lower levels of UA and U, the main antioxidants in the serum, are associated with an increased risk of developing PD, while higher levels (including those above the upper reference range), appear to be protective, despite increasing the risk for other diseases—for details concerning the proposed mechanisms, see Figure 2. Our study adds to the evidence that lower serum UA levels are significantly associated with the severity of motor impairment in PD (i.e., the severity of parkinsonism assessed through the Hoehn and Yahr scale), as well as with the existence of motor complications of long-term dopaminergic treatment, and with neurocognitive impairment. Although in our study serum UA levels were also associated with the male sex, we found no correlation between sex and motor features of PD or neurocognitive impairment. Additionally, we found no significant associations with disease duration or with the dopaminergic medication.

The illustration above describes the mechanisms by which oxidative stress is involved in the onset and progression of PD. Disruption of striatal dopamine metabolism by oxidation, mitochondrial dysfunction, excitotoxicity, and apoptosis are consequences of excessive free radicals and may alter neuronal physiology. UA is a powerful antioxidant which, in addition to removing oxygen free radicals, is iron chelating, diminishing the oxidizing potential of Fe^3+^.

Our findings are in line with pre-existing evidence indicating that lower UA and U levels may be associated with more severe parkinsonism. The available literature suggests that lower UA and U levels may also associate with other clinical features, such as postural instability and gait impairment [45,46], as well as depression, neurocognitive impairment, and other non-motor symptoms of PD [28,47,48]. There is also evidence suggesting a relationship between higher serum U levels and milder motor phenotypes [49]. Three meta-analyses showed lower levels of serum UA in patients with PD compared with healthy controls [50,51,52], a potentially protective effect of higher serum UA levels [52], and a decrease in serum UA levels as PD progresses [51]. Lower levels of serum UA might influence the subsequent development of mild cognitive impairment. Nevertheless, a meta-analysis on UA and non-motor features of PD, including neurocognitive impairment, did not prove an association, warranting the need for further studies [53]. UA might be excessively consumed in PD, a process putatively amplified by dopaminergic medication. This could explain the lower serum UA levels in more advanced PD and in patients with a higher daily levodopa intake [54]. Necroptic studies found decreased UA levels in the substantia nigra of patients with PD, which could also suggest increased consumption, related to the excessive oxidative stress caused by local dopamine [13]. Although these hypotheses remain to be proven, they support a potential therapeutic role of UA and U in PD. As previously mentioned, two recent population-based studies looked at the link between gout, a condition characterized by high UA, and PD, finding no association between the two [39,40]. One hypothesis explaining this finding is that the proinflammatory status of gout might result in excessive oxidative stress that counteracts the potentially beneficial antioxidant effects of the higher levels of UA. A potential role played by the anti-gout medication, which lowers UA, has also been suggested.

Though the abovementioned associations do not prove causation, they are biologically plausible. UA has neuroprotective effects in experimental PD models [55], partly by modulating oxidative stress and neuroinflammation through the Nrf2-ARE pathway [56]. On the other hand, hyperuricemia has been linked with intestinal barrier disfunction, dysregulated intestinal immunity, and systemic inflammation, which may also play a role in PD [57]. Recently, gut dysbiosis has been identified as a potentially important environmental factor that may promote alpha-synuclein expression and aggregation in the enteric neurons, and may alter the intestinal barrier permeability, resulting in increased risk of neuronal exposure to various microbial or nonmicrobial xenobiotics, with direct or indirect proinflammatory or neurotoxic effects. Interestingly, experimental data on a rat model indicate that the composition of gut microbiota is associated with hyperuricemia [58]. This highlights the importance of caution in any attempts to raise serum UA levels.

Our study has a series of limitations, the first being the small sample size. Another important limitation is the medical record-based observational design, which implies a non-standardized collection of data and limited possibility to evaluate the associations with some characteristics of the disease or with other factors of interest (such as dietary habits or other potential confounding factors). Moreover, the cross-sectional design means that data on the evolution of the disease are only in small part based on previous medical records, most being subjected to recollection bias. Additionally, our study did not include data from patients with probable monogenic PD (as genetic testing was not performed in most patients). Interestingly, higher levels of serum UA are associated with a higher risk of developing PD in carriers of pathologic variants of the leucine-rich repeat kinase 2 (LRRK2) gene, the most common cause of familial PD [59], and these patients have lower levels of serum UA compared to healthy controls [60]. A significant association was also found between lower serum UA levels and the GBA1 mutation PD [61].

Despite the above limitations, our findings are important because they add to the evidence that serum UA levels are associated with motor and neurocognitive impairment in patients with PD. This means that serum UA and U deserve further research, both as biomarkers for PD progression, and as plausible therapeutic targets for disease-modyfing interventions. In this respect, serum UA and U levels can be influenced by means of diet; for instance, a purine rich diet (i.e., rich in meat products) would increase their serum levels [62]. Intravenous administration of U is well tolerated and significantly increases the total antioxidant capacity of the serum [63], but is impractical. As oral U has low bioavailability, its precursor, inosine, may be administered to increases U levels both in the serum and in the cerebrospinal fluid. A randomized, double-blinded, placebo-controlled safety, tolerability, and dose-ranging trial of oral inosine in PD (i.e., Safety of Urate Elevation in PD, SURE-PD) found that oral inosine titrated to mildly or moderately elevate the serum U levels is safe and well tolerated for up to 2 years (except for the development of kidney stones in 6% of those receiving inosine), leads to an increase in the cerebrospinal fluid concentration of U, and may show a trend towards a modest reduction in PD progression, especially in women [64]. Despite its good safety profile, a subsequent inosine trial, Effect of Urate-Elevating Inosine on Early Parkinson Disease Progression (SURE-PD3), showed no significant differences between the active and the placebo arms concerning the progression of PD [65].

## 5. Conclusions

The serum UA level is a potential marker for the risk of PD development and progression and might provide the opportunity for potential therapeutic interventions for prodromal PD. Considering the current lack of prophylactic or disease-modyfing therapies in PD, and the increasing epidemiological and experimental evidence in support of a prospective role of UA and U in PD, further reasearch is warranted.

## Figures and Tables

**Figure 1 medicina-58-00245-f001:**
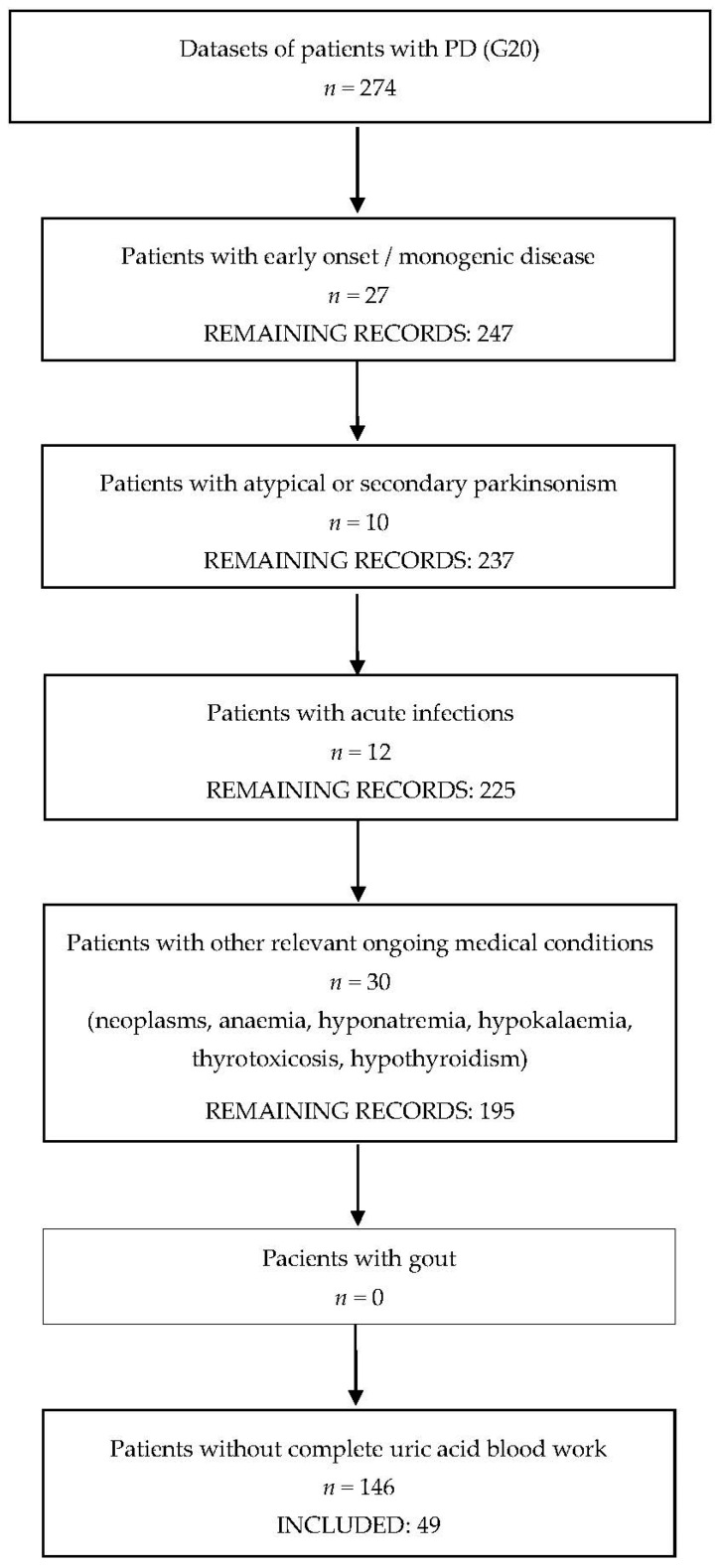
The inclusion process of the study.

**Figure 2 medicina-58-00245-f002:**
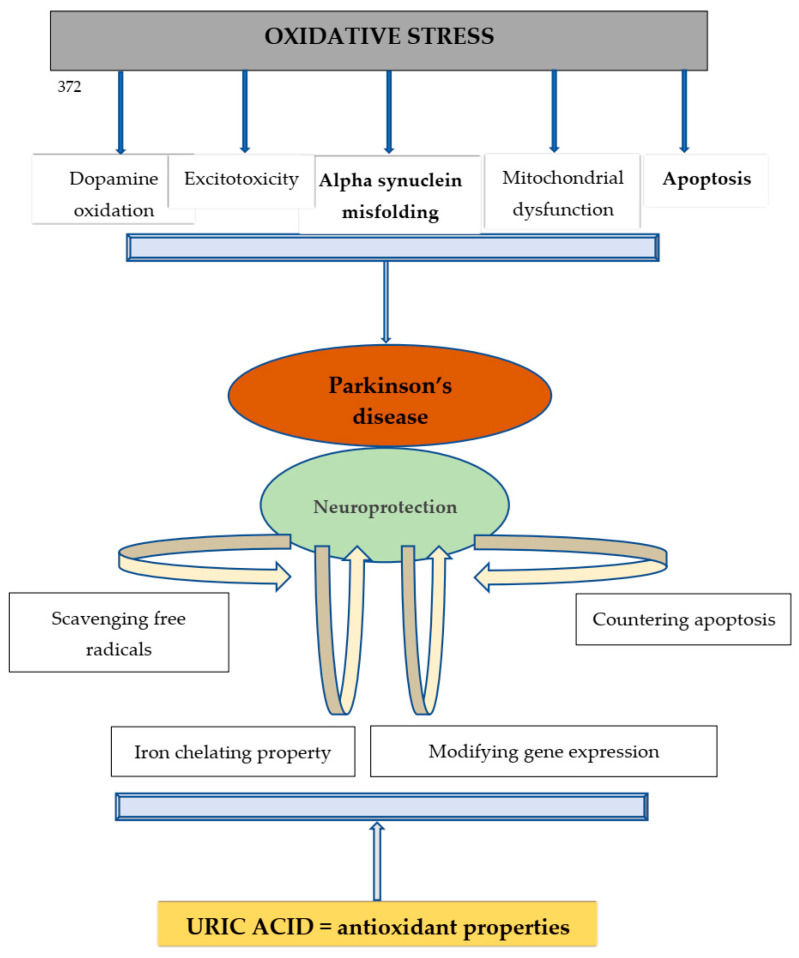
Mechanisms of oxidative stress in the pathogenesis of PD and UA-related mechanisms that may contribute to neuroprotection.

**Table 1 medicina-58-00245-t001:** Demographic and disease characteristics of the patients with PD included in the study.

Characteristic	Result (*n*, %)
Sex (M: F)	33 (67.3%): 16 (32.7%)
Mean age (years)	69.1 ± 2.4, SD 8.6
Age 65 years and older	36 (73.5%)
Mean disease duration (years)	5.1 ± 1.8, SD 5.2
Hoehn and Yahr scale ^1^	Stage ≤ 2.5: 22 (44.9%) Stage ≥ 3: 18 (36.7%)
Motor complications	14 (28.6%)
Neurocognitive impairment	16 (32.7%)
Cerebrovascular disease	13 (26.5%)
Mean daily levodopa equivalent dose (mg)	925.2 ± 178.2, SD 603.1

^1^ missing data 18.4%. Abbreviations: F = female, M = male, *n* = number, PD = Parkinson’s disease, SD = standard deviation.

**Table 2 medicina-58-00245-t002:** Associations between the PD characteristics and lower UA levels in the study population.

Associations of Lower UA Levels	Correlation Coefficient	*p*-Value
Severity of parkinsonism (Hoehn and Yahr scale)	r_s_ = 0.488	0.002
Motor complications	r = 0.333	0.027
Neurocognitive impairment	r = 0.346	0.021

Abbreviations: PD = Parkinson’s disease, r = Pearson coefficient with point-biserial correlations, r_s_ = Spearman correlation coefficient, UA = uric acid.

## Data Availability

Anonymized data will be made available upon reasonable request made to the corresponding authors.

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
