# Peer review of "Serum Uric Acid Levels in Parkinson’s Disease: A Cross-Sectional Electronic Medical Record Database Study from a Tertiary Referral Centre in Romania"

_medicina, 2022, doi:10.3390/medicina58020245_

Round 1

Reviewer 1 Report

1) The study type should be mentioned in the title.

2) It is advised to review grammatical English throughout the text. There are significant grammatical errors and redundant phrases.

3) Introduction should be revised. Many references are grouped. It is advised to choose only one. E.g.: ''Epidemiological studies suggest that lower levels of serum UA and U are associated with an increased risk of PD [20-26]'' ''UA levels possibly also influencing the risk of PD [32-34]''

4) Could the author provide the spreadsheet of the cases as supplementary material? ‘‘All personal data were anonymized, as per the General Data Protection Regulation’’

5) Please provide funding details if available. ‘‘The study was conducted as part of the doctoral research of the first author’’

6) Methods

i) Explain ‘‘Hippocrate Electronic Medical Record software’’

ii) Were at least two board-certified neurologists that did the diagnosis of PD?

iii) Were smoking individuals excluded?

iv) How did you analyze if the distribution of your data?

v) How the power of the study was calculated?

vi) Could the authors provide Spearman correlation graphs as supplementary material

vii) What was the SPSS edition used?

7) Results. Medications in use to manage PD should be further described. Were all of the individuals were only using levodopa?

8) Discussion should discuss the data of the study with the present literature.

9) References should be updated. Only 43% (38/88) is from the last five years of science. It is advised > 70%.

NEW IDEAS

A) It is advised to do logistic regression with the data

B) Provide spreadsheet data

C) Figure merging pathophysiological mechanisms and data of the present manuscript.

D) Discuss the contradiction of already two great studies already reported within the last two years.

  • - Kim JH, Choi IA, Kim A, Kang G. Clinical Association between Gout and Parkinson's Disease: A Nationwide Population-Based Cohort Study in Korea. Medicina (Kaunas). 2021 Nov 24;57(12):1292. doi: 10.3390/medicina57121292. PMID: 34946237; PMCID: PMC8704991.
  • Hu LY, Yang AC, Lee SC, You ZH, Tsai SJ, Hu CK, Shen CC. Risk of Parkinson's disease following gout: a population-based retrospective cohort study in Taiwan. BMC Neurol. 2020 Sep 8;20(1):338. doi: 10.1186/s12883-020-01916-9. PMID: 32900384; PMCID: PMC7487828.

-----------------------------------------------------------------

It is advised to the author think in this phrase ‘‘What does this manuscript add new for the present literature?’’

Author Response

We thank the reviewers for their constructive comments and recommendations that helped improve our manuscript. Please find below our responses, point by point. 

Reviewer 1:

Thank you for your thorough review, your comments and recommendations have been very helpful in improving our manuscript. 

1) The study type should be mentioned in the title.

Thank you. We changed the title accordingly. 

2) It is advised to review grammatical English throughout the text. There are significant grammatical errors and redundant phrases.

As suggested, we revised the English grammar, and we changed the redundant phrases.

3) Introduction should be revised. Many references are grouped. It is advised to choose only one. E.g.: ''Epidemiological studies suggest that lower levels of serum UA and U are associated with an increased risk of PD [20-26]'' ''UA levels possibly also influencing the risk of PD [32-34]''

Thank you for this suggestion. The introduction has been revised so that references are not grouped anymore – however, since all except for one reference were to relevant studies, we kept most of them.

4) Could the author provide the spreadsheet of the cases as supplementary material? ‘‘All personal data were anonymized, as per the General Data Protection Regulation’’

A spreadsheet of the cases was added as supplementary material – please see Table 1 and Table 2.

5) Please provide funding details if available. ‘‘The study was conducted as part of the doctoral research of the first author’’

Funding has been available only for publication fees. The funding details have been added in the funding section. Thank you.

6) Methods

  1. i) Explain ‘‘Hippocrate Electronic Medical Record software’’

The Hippocrate Electronic Medical Record software is the program used at Colentina Clinical Hospital (since March 2018) for the digital recording / storage of medical information. It is used in hospitals in over 133 countries around the world. A brief explanation has been added to the text.

  1. ii) Were at least two board-certified neurologists that did the diagnosis of PD?

This was a study performed on electronic medical records, so this does not apply. However, the diagnosis of PD was made by a neurologist working at Colentina Clinical Hospital, which is a tertiary referral centre for PD in Romania. Moreover, difficult cases are routinely discussed with the corresponding author, who is an internationally renowned neurologists and professor of neurology with high expertise in the field of PD and neurodegenerative disorders. A brief clarification has been added to the text.

iii) Were smoking individuals excluded?

No. Indeed this would have been interesting to check, but not enough data was available to accurately exclude smokers.

  1. iv) How did you analyze if the distribution of your data?

Normal distribution of data was analyzed using the Shapiro-Wilk test, in SPSS Subscription. This has been added to the text.

  1. v) How the power of the study was calculated?

The study included all the available data from the electronic medical records, so sample size was not calculated prior to data inclusion. We performed a post hoc power for the Hoehn and Yahr stage, motor complications and neurocognitive impairment and mentioned it in the text.

  1. vi) Could the authors provide Spearman correlation graphs as supplementary material

We added the Spearman correlation graphs as supplementary material.

vii) What was the SPSS edition used?

We used SPSS Subscription (Subscription is updated in real time and does not have an edition). SPSS Subscription is mentioned in the text. 

7) Results. Medications in use to manage PD should be further described. Were all of the individuals were only using levodopa?

Most patients were using levodopa, but not all. We added details on the medication (including LEDD) in the supplementary materials.

8) Discussion should discuss the data of the study with the present literature.

Thank you for suggesting this. We expanded the discussion as advised.

9) References should be updated. Only 43% (38/88) is from the last five years of science. It is advised > 70%.

We completely agree. However uric acid in PD has not been a very hot topic in recent years, so we consider that older papers are still relevant. We searched the literature once again and included 2 more relevant papers from the past two years; also, we removed a 1981 paper on uric acid, which could have been considered redundant in the light of more recent data, as well as 3 other older papers that could have been consider redundant.

NEW IDEAS

  1. A) It is advised to do logistic regression with the data

This was performed in SPSS Subscription.

  1. B) Provide spreadsheet data

We provided the spreadsheet of cases in the supplementary material.

  1. C) Figure merging pathophysiological mechanisms and data of the present manuscript.

We added Fig. 2, in which we provided this.

  1. D) Discuss the contradiction of already two great studies already reported within the last two years.
  • - Kim JH, Choi IA, Kim A, Kang G. Clinical Association between Gout and Parkinson's Disease: A Nationwide Population-Based Cohort Study in Korea. Medicina (Kaunas). 2021 Nov 24;57(12):1292. doi: 10.3390/medicina57121292. PMID: 34946237; PMCID: PMC8704991.
  • Hu LY, Yang AC, Lee SC, You ZH, Tsai SJ, Hu CK, Shen CC. Risk of Parkinson's disease following gout: a population-based retrospective cohort study in Taiwan. BMC Neurol. 2020 Sep 8;20(1):338. doi: 10.1186/s12883-020-01916-9. PMID: 32900384; PMCID: PMC7487828.

We dealt with this in the Discussion section, and added the suggested articles as references (Hu et al was already included).

It is advised to the author think in this phrase ‘‘What does this manuscript add new for the present literature?’’

We considered this in the Discussion section.

Reviewer 2 Report

The authors reported an interesting and well written study about serum uric acid levels in PD. I have some comments to the authors:

1) In the first paragraph of the introduction (lines 32-34) the authors should split the sentences "It is responsible for progressive motor... " in two: one dedicated to motor symptoms and the other to non-motors symptoms. In the motor symptoms part it is not sufficient saying "Parkinsonism", please include tremor, bradykinesia, rigidity etc. As far as non-motor symptoms are concerned please add in the list of symptoms "hyposmia, anosmia, dysgeusia, olfactory and gustatory hallucinations, sleep disorders". I suggest these references in order to offer to the readers a wider view on this topic: Moustafa AA, et al. Motor symptoms in Parkinson's disease: A unified framework. Neurosci Biobehav Rev. 2016 Sep;68:727-740. doi: 10.1016/j.neubiorev.2016.07.010. Epub 2016 Jul 12. PMID: 27422450.
Schapira AHV, et al. Non-motor features of Parkinson disease. Nat Rev Neurosci. 2017 Jul;18(7):435-450. doi: 10.1038/nrn.2017.62.
Solla P, et al. Frequency and Determinants of Olfactory Hallucinations in Parkinson's Disease Patients. Brain Sci. 2021 Jun 24;11(7):841. doi: 10.3390/brainsci11070841. PMID: 34202903; PMCID: PMC8301996.

2) In figure 1 please specify that you have excluded patients suffering from gout.

3) please include among the limitations the small sample size.

Author Response

We thank the reviewers for their constructive comments and recommendations that helped improve our manuscript. Please find below our responses, point by point. 

Reviewer 2:

Thank you for the kind appraisal and the constructive recommendations that have helped us to improve our manuscript.

The authors reported an interesting and well written study about serum uric acid levels in PD. I have some comments to the authors:

  • In the first paragraph of the introduction (lines 32-34) the authors should split the sentences "It is responsible for progressive motor... " in two: one dedicated to motor symptoms and the other to non-motors symptoms. In the motor symptoms part it is not sufficient saying "Parkinsonism", please include tremor, bradykinesia, rigidity etc. As far as non-motor symptoms are concerned please add in the list of symptoms "hyposmia, anosmia, dysgeusia, olfactory and gustatory hallucinations, sleep disorders". I suggest these references in order to offer to the readers a wider view on this topic: Moustafa AA, et al. Motor symptoms in Parkinson's disease: A unified framework. Neurosci Biobehav Rev. 2016 Sep;68:727-740. doi: 10.1016/j.neubiorev.2016.07.010. Epub 2016 Jul 12. PMID: 27422450.
    Schapira AHV, et al. Non-motor features of Parkinson disease. Nat Rev Neurosci. 2017 Jul;18(7):435-450. doi: 10.1038/nrn.2017.62.
    Solla P, et al. Frequency and Determinants of Olfactory Hallucinations in Parkinson's Disease Patients. Brain Sci. 2021 Jun 24;11(7):841. doi: 10.3390/brainsci11070841. PMID: 34202903; PMCID: PMC8301996.

Thank you for suggesting this. We have expanded the notion of "parkinsonism", as suggested, and added the proposed non-motor symptoms. Also, we have included the recommended bibliography.

2) In figure 1 please specify that you have excluded patients suffering from gout.

We have included this specification in Figure 1.

3) please include among the limitations the small sample size.

We have included the small sample size among the limitations.

Round 2

Reviewer 1 Report

1) Some grammatical errors throughout the text should be addressed. Punctuation, long phrases. It is advised a professional editing service.
2) Methods
a) cross-sectional observational study = cross-sectional
b) correlations = association
c) ''A summary of the results has already been presented as an electronic poster at the 15th World Congress on Controverses in Neurology (P-110 / 2021).'' It is advised to remove from methods and include this structure in the disclosure section.
d)  IBM Statistical Package for Social Sciences (SPSS). Which version of SPSS was used?
e) If the distribution was normal. Why did you use the Spearman correlation coefficient?
https://stats.stackexchange.com/questions/3730/pearsons-or-spearmans-correlation-with-non-normal-data
f) The authors need to describe the statical methods better. For example, why such statistical tests were used and for which variables? How were the parameters chosen for logistic regression? Why did you use only the Shapiro-Wilk test to the analysis of the distribution of the data?
https://statistics.laerd.com/spss-tutorials/testing-for-normality-using-spss-statistics.php
g) A table is needed to display the results of logistic regression. At least the confidence interval, unstandardized regression weight, p-value, and the measurement of likelihood must be presented.
3) References section should be revised. It is advised no more than 50 references. The authors should revise the text for grouped references. Eg.: ''... proinflammatory or neurotoxic effects [71-79]''

The authors should include the information requested by the reviewer, in the first round, on the manuscript.

Author Response

We do thank indeed the reviewer for the comments and recommendations that helped improve our manuscript. Please find our answers below.

  • Some grammatical errors throughout the text should be addressed. Punctuation, long phrases. It is advised a professional editing service.

We thank the reviewer for pointing this out, indeed, the text needed to be improved. We corrected the grammatical and punctuation errors and shortened the long phrases.

2) Methods
a) cross-sectional observational study = cross-sectional

Thank you, we made the correction.

  1. b) correlations = association

Thank you, we made the correction.

  1. c) ''A summary of the results has already been presented as an electronic poster at the 15th World Congress on Controverses in Neurology (P-110 / 2021).'' It is advised to remove from methods and include this structure in the disclosure section.

We thank the reviewer for their advice. We changed the text accordingly.

  1. d) IBM Statistical Package for Social Sciences (SPSS). Which version of SPSS was used?

The version we used is SPSS Statistics Subscription (we clarified this in the text). SPSS Subscription is a subscription-based plan which does not have an edition number, contrary to the traditional licence-based service. Please see: https://www.ibm.com/downloads/cas/YDWQGBDL .

  1. e) If the distribution was normal. Why did you use the Spearman correlation coefficient?
    https://stats.stackexchange.com/questions/3730/pearsons-or-spearmans-correlation-with-non-normal-data

For the distribution, please see Supplementary Table 4 – the distribution was normal for age and uric acid levels.

We thank the reviewer for pointing out the issue with using the Spearman correlation coefficient for categorical dichotomous variables. We corrected this, replacing the results of the Spearman correlation coefficient with the Pearson coefficient with point-biserial correlation. Please see the text and the Supplementary Figures 10 and 11.

  1. f) The authors need to describe the statical methods better. For example, why such statistical tests were used and for which variables? How were the parameters chosen for logistic regression? Why did you use only the Shapiro-Wilk test to the analysis of the distribution of the data?

https://statistics.laerd.com/spss-tutorials/testing-for-normality-using-spss-statistics.php

We thank the reviewer for the advice. We described the statistical methods better.

We performed logistic regression to evaluate the relationship between serum uric acid level and presence or absence of motor complications and cognitive impairment. Therefore, we chose serum uric acid level as the independent variable (a continuous, ratio variable), and motor complications and cognitive impairment, as dependent variables (both categorical, dichotomous variables).

The Shapiro-Wilk test was not the only means used to test distribution, but it was the only one mentioned in the manuscript, since it is more appropriate for small sample sized studies (i.e., under 50), such as ours. We analysed the distribution of data in SPSS Subscription, for the variables age and serum uric acid level, using both numerical and graphical methods. Another test of normality was the Kolmogorov-Smirnov test, and the graphs were the histogram, the normal Q-Q plot, detrended normal Q-Q plot and observed value for age and serum uric acid, respectively (we have added the table with tests of normality, namely Kolmogorov-Smirnov and Shapiro-Wilk, and the graphs, in the supplementary material). We briefly clarified this in the text.

  1. g) A table is needed to display the results of logistic regression. At least the confidence interval, unstandardized regression weight, p-value, and the measurement of likelihood must be presented.

We have provided the table with the results of logistic regression, including the required data, in the supplementary material.

3) References section should be revised. It is advised no more than 50 references. The authors should revise the text for grouped references. Eg.: ''... proinflammatory or neurotoxic effects [71-79]''

Thank you. We made the changes, as suggested, and removed 12 references. Although the manuscript still has 70 references, we find it difficult to remove any more references without removing information that is relevant for the topic.

The authors should include the information requested by the reviewer, in the first round, on the manuscript.

This has been done in the initial revision. We have replaced the Spearman graphs accordingly.